# The Accuracy of the Uganda National Tuberculosis and Leprosy Program diagnostic algorithm and the World Health Organisation treatment decision algorithms for childhood tuberculosis: A retrospective analysis

Peter J. Kitonsa[1,2]*, Bernard Kikaire[2], Peter Wambi[1], Annet Nalutaaya[1], Jascent Nakafeero[1], Gertrude Nanyonga[1], Emma Kiconco[1], Deus Atwiine[1,3], Robert Castro[4], Ernest A. Oumo[1], Hellen T. Aanyu[3], Mary N. Mudiope[5], Ezekiel Mupere[2], Moorine P. Sekadde[6], Swomitra Mohanty[7], Adithya Cattamanchi[1,4,8,9], Eric Wobudeya[1,3], Devan Jaganath[1,9,10]

1 Uganda Tuberculosis Implementation Research Consortium (U-TIRC), Walimu, Kampala, Uganda, 2 Department of Pediatrics and Child Health, College of Health Sciences, Makerere University, Kampala, Uganda, 3 Department of Pediatrics and Child Health, Mulago National Referral Hospital, Kampala, Uganda, 4 Division of Pulmonary and Critical Care Medicine, University of California, San Francisco, United States of America, 5 Infectious Diseases Institute, Kampala, Uganda, 6 National TB and Leprosy Programme (NTLP), Ministry of Health, Kampala, Uganda, 7 Departments of Chemical and Metallurgical Engineering, University of Utah, Salt Lake City, United States of America, 8 Division of Pulmonary Diseases and Critical Care Medicine, University of California, Irvine, United States of America, 9 Center for Tuberculosis, Institute for Global Health Sciences, University of California, San Francisco, United States of America, 10 Division of Pediatric Infectious Diseases, University of California, San Francisco, United States of America

* kitonsap@gmail.com

## Abstract

Diagnosing childhood pulmonary tuberculosis (TB) is a challenge. This led the Uganda National Tuberculosis and Leprosy Program (NTLP) to develop a clinical treatment decision algorithm (TDA) for children. However, there is limited data on its accuracy, and how it compares to new World Health Organization (WHO) TB TDAs for children. This study aimed to evaluate and compare the accuracy of the 2017 Uganda NTLP diagnostic algorithm with the 2022 WHO TDAs for TB among children. We analyzed four years of clinical data from children <15 years old in Kampala, Uganda. Children were classified as per National Institutes of Health (NIH) consensus definitions (Confirmed, Unconfirmed or Unlikely TB). We applied the 2017 Uganda NTLP and 2022 WHO algorithms (A with chest x-ray [CXR], B without CXR) to make a decision to treat for TB or not, and calculated the sensitivity, specificity and predictive values in reference to Confirmed vs. Unlikely TB, as well as a microbiological and composite reference standard. Of the 699 children included in this analysis, 64% (451/699) were under 5 years, 53% (373/669) were male, 12% (85/699) were Xpert Ultra positive, 11% (74/669) were HIV positive and 6% had severe acute malnutrition (SAM). The Uganda NTLP algorithm had a sensitivity of 97.9% (95% CI: 96.4-99.4) and specificity of 25.9% (95% CI: 21.2-30.7). If CXR was considered

**Data availability statement:** Data set uploaded as Supplementary Material.

**Funding:** The study was supported by the National Institutes of Health (R01HL139717 and U01AI152087 to AC, and R01HL169449 and K23HL153581 to DJ). The funders had no role in study design, data collection and analysis, decision to publish, or preparation of the manuscript.

**Competing interests:** The authors have declared that no competing interests exist.

unavailable, sensitivity was 97.9% (95% CI: 96.4-99.4) and specificity 28.1% (95% CI: 23.2-33.0). In comparison, WHO TDAs had similar sensitivity to the Uganda NTLP, but algorithm A was more specific (32.2%, 95% CI: 26.9-37.5) and algorithm B was less specific (15.4%, 95% CI: 11.3-19.5). The WHO TDAs had better specificity than the NTLP algorithm with CXR, and worse specificity without CXR. Further optimization of the algorithms is needed to improve specificity and reduce over-treatment of TB in children.

## Author summary

In this study, we assessed and compared the accuracy of the 2017 Uganda National TB and Leprosy Program (NTLP) algorithm with the 2022 World Health Organization (WHO) Treatment Decision Algorithms (TDAs) among children evaluated for pulmonary tuberculosis (TB) in Kampala, Uganda. Using retrospective data from 699 children under 15 years old, we determined the sensitivity, specificity, and predictive values of these algorithms for pediatric TB. Our findings show that while both the Uganda NTLP and WHO TDAs had high sensitivity, their specificity remained low. The WHO TDA with chest X-ray (CXR) demonstrated better specificity than the Uganda NTLP algorithm, whereas the WHO TDA without CXR had lower specificity. These results highlight the need for further optimization of TB diagnostic algorithms to improve specificity and reduce unnecessary treatment in children. Strengthening diagnostic accuracy is crucial for better TB management and resource allocation in high-burden settings.

## Introduction

Childhood pulmonary tuberculosis (TB) presents a diagnostic challenge due to its non-specific presentation, and difficulties in obtaining respiratory specimens in young children. Even if sputum or stool is obtained, molecular testing is often negative due to the paucibacillary nature of childhood TB [1,2]. Moreover, children often present at primary care facilities, which may lack the infrastructure for molecular testing or chest X-ray (CXR) to support TB evaluation and timely diagnosis. Consequently, there are large delays in TB treatment [3], which may explain why globally treatment coverage was only 55% for children < 15 years (and even lower for the age group below 5 years at 48%) according to the Global TB report of 2024 [1].

To reduce the delays in TB treatment initiation for children, treatment decision algorithms (TDAs) have been developed to guide providers on empiric treatment based on signs and symptoms. In Uganda, the National TB and Leprosy Program (NTLP) developed an algorithm that has been implemented since 2017 [4]. In 2022, the World Health Organization (WHO) conditionally recommended the use of TDAs for childhood TB, and presented (in its operational handbook) an algorithm based on signs and symptoms as well as scores with CXR (Algorithm A) or without CXR (Algorithm B) [5]. However, these TDAs have only been internally validated, and therefore the WHO has called for their external validation. Moreover, it is unclear how the new WHO algorithms would perform in Uganda in comparison to the current Uganda NTLP algorithm.

To better define the role of these algorithms, we retrospectively analyzed four years of clinical data from children in Kampala, Uganda who underwent a standard TB evaluation and were classified according to National Institutes of Health (NIH) consensus definitions for childhood TB [6]. We utilized this data to determine the accuracy of the Uganda NTLP

algorithm, and also to compare its accuracy it to the WHO TDAs in same age group, and how accuracy varied based on access to Xpert and/or CXR, and TB prevalence.

## Materials and methods

### Study population

We performed a secondary analysis of data from children aged 0 - 14 years who had undergone an evaluation for possible pulmonary TB between 28th November 2018 and 30th November 2022. The parent study, whose main aim was to evaluate performance of novel diagnostic tests, was situated at the Mulago National Referral Hospital Pediatric TB clinic within the Mulago National Referral Hospital (MNRH) [7]. The TB clinic recruited children from the MNRH pediatric wards, Mulago assessment center, other lower level clinics and hospitals in and around Kampala (including Kampala Capital City Council (KCCA) clinics/health facilities and Infectious Diseases Institute (IDI) clinics), and referrals from upcountry health facilities. After evaluation, children returned to their referring facilities for further care. MNRH provides approximately 500,000 outpatient visits per year including from specialized clinics [7]. The hospital's immediate catchment area includes Kampala, Mukono and Wakiso districts with a population of approximately 1.8, 0.9 million and 3.4 million respectively [8]. The main languages spoken are Luganda and English. The hospital is also a national center of excellence that attends to patients from the entire country.

The parent study consecutively enrolled children if they had cough at least 1 week and at least two of the following: unexplained fever >1 week, weight loss (>5% reduction compared to highest recorded weight in prior 3 months), poor appetite, poor weight gain/failure to thrive (weight-for-age or weight-for-height/length z-scores ≤-3), unexplained lethargy or reduced playfulness >1week, abnormal chest x-ray or history of TB contact. Children who were already on TB treatment, or history of TB disease in the last year, were excluded. Our analysis was restricted to children who met the Uganda intensified TB case finding (ICF) guide [4] definition of presumptive TB applied at all entry points: any one of persistent cough ≥2 weeks or any duration if HIV positive: persistent fevers ≥2 weeks: poor weight gain in the last ≥1 month; TB contact, or swellings in the neck, armpit or groin. The data from the parent study was not used to develop the WHO TDAs.

### TB evaluation procedures

Participants completed an interview, which included a standardized questionnaire about TB symptoms and signs and the duration of each, along with questions about their sociodemographic information and potential TB risk factors. The clinical evaluation included physical examination, CXR, Tuberculin skin testing (TST), and HIV testing. Respiratory specimen collection included expectorated or induced sputum, gastric aspiration, or nasopharyngeal aspiration, and was sent for Gene Xpert MTB/RIF Ultra testing [Cepheid, Sunnyvale, USA] and solid or liquid mycobacterial culture. Xpert Ultra trace semi-quantitative results were considered positive. Additional details have been previously published [9].

All individuals had their weight (in kg) and height/length (in meters) taken using a weighing scale and SECA-216 stadiometer (Seca Industries, Hamburg), respectively. All children had a follow up visit regardless of TB diagnosis at 2 months to assess ongoing or resolved signs and symptoms and any response to anti-TB treatment.

### Classification of TB status by treatment decision algorithms

According to the NTLP algorithm [4], if a child < 15 years old meets ICF criteria for presumptive TB, it is recommended to obtain a respiratory specimen for molecular testing with Xpert MTB/

RIF Ultra. If results show MTB detected (including trace), they are treated for TB. If MTB is not detected, the decision to treat is based on whether they meet ≥2 of the following criteria if HIV negative OR ≥1 criteria if HIV positive; 1. two or more TB symptoms (persistent fever≥2 weeks, persistent cough≥2 weeks, poor weight gain ≥1 month), 2. positive history of TB contact (bacteriologically confirmed), 3. CXR suggestive of TB, and 4. any physical signs suggestive of TB. Physical signs suggestive of TB include: severe malnutrition, cervical, axillary or groin lymphadenopathy, features of pneumonia not responding to broad spectrum antibiotics (S1 Appendix) [10].

The 2022 WHO TB treatment decision algorithms (TDAs) A and B were developed for children <10 years old, and so analysis was limited to this age group [11]. The algorithms both recommend molecular testing for children with presumptive pulmonary TB. If negative or unavailable, a TB score is calculated that assigns points to TB signs and symptoms, with TB treatment decision when the total score ≥10. The algorithms differ in whether CXR is available (A) or is not (B), with corresponding different scores assigned to the TB features (S2 Appendix and S3 Appendix) [12].

We analyzed TB treatment decisions for four scenarios: (1) when Xpert and CXR is available; (2) when only CXR is available; (3) when only Xpert is available; and (4) when neither Xpert nor CXR is available. The Uganda NTLP algorithm was also evaluated for all the four scenarios, and WHO TDA A for (1) and (2), and TDA B for (3) and (4).

## Reference standard

Children were classified as Confirmed TB (positive Xpert or culture), Unconfirmed TB (negative microbiological testing but consideration of other signs and symptoms of TB and response to TB treatment) or Unlikely TB (not meeting other criteria) using the NIH consensus definitions [6]. They were unclassifiable if there was insufficient data to determine TB status. Two pediatric TB experts classified each participant independently and blinded to the TDA calculation, with a third reviewer as a tiebreaker if needed. We used the following reference standards:

1. **Confirmed TB versus Unlikely TB**. This excludes children with Unconfirmed TB.

2. **Composite reference standard (CRS)**. Confirmed or Unconfirmed are defined as TB, and Unlikely as not having TB.

3. **Microbiologic reference standard (MRS)**. Confirmed TB was defined as TB, and Unconfirmed or Unlikely TB as not having TB.

## Statistical methods

Descriptive statistics were computed to describe relevant characteristics, and categorical variables were summarized using frequencies and proportions and all continuous variables were summarized using medians and interquartile range (IQR). Our primary objective was to determine the accuracy of the algorithms. Utilizing each reference standard above, we calculated the sensitivity and specificity of the Uganda NTLP algorithm (children <15years) and the WHO TDAs (children <10 years) for the overall estimate and for key subgroups of children <5 years old, with SAM, or living with HIV; these were defined as high-risk by the WHO due to their rapid disease progression. We calculated the positive predictive value (PPV) and negative predictive value (NPV) under three TB prevalence estimates (5%, the sample prevalence, and 20%) using the conditional probabilities method [13]. We further compared the accuracy of the Uganda NTLP algorithm with that of the WHO TDAs, among children <10 years. We compared the sensitivity and specificity of each algorithm using McNemar's test, with significance defined as p-value < 0.05.

Stata version 13.0 (Statacorp, College Station, USA) was used for this analysis [14]. We followed the Standards for Reporting Diagnostic Accuracy (STARD) guidelines for this study (S4 appendix) [15].

### Ethical considerations

The study was approved by the Mulago Hospital Research and Ethics Committee, Kampala, Uganda (IRB number 12884), and the institutional review board at the University of California, San Francisco (UCSF), United States. Caregivers provided informed written consent and capable children ≥8 years provided assent for all study activities.

## Results

### Participant characteristics and TB

Between November 2018 to November 2022, there was a total of 875 children with TB symptoms of whom 867 met the intensified TB case finding (ICF) criteria, and of whom 842 were enrolled into the study (Fig 1). TB status was unclassifiable for 143 participants and 25 did not consent; and were excluded. We thus analyzed data from a total of 699 children, of whom 98 (14.1%) had Confirmed TB, 361 (51.6%) had Unconfirmed TB, and 240 (34.3%) had Unlikely TB.

The participants' median age was 3 years (Interquartile Range, IQR: 1-6), 53% (373/699) were male, 64% (451/699) were aged <5 years, 92% (642/699) were aged <10 years, 57% (399/699) had history of TB contact, and 6% (42/699) had severe acute malnutrition (SAM). There were 74 (11%) children with HIV (CHIV), and 12% (85/699) of children had a positive Xpert result (Table 1).

### Diagnostic accuracy of the Uganda NTLP algorithm

When CXR was available, the overall sensitivity was 97.9% (95% CI: 96.4-99.4) among Confirmed vs. Unlikely TB and the MRS, and 90.7% (95% CI: 88.5-92.9) with the CRS. Specificity was 25.9% (95% CI: 21.2-30.7) in reference to Confirmed vs. Unlikely TB and the CRS, and 17.1% (95% CI: 14.3-20.0) with the MRS (Table 2).

When CXR was unavailable, the sensitivity remained the same in reference to Confirmed vs. Unlikely TB and the MRS, and reduced modestly with the CRS from 90.7% to 88.4%. In contrast, there were small increases in specificity and ranged from 19.7% to 28.1% (Table 2). At TB prevalences of 5%, 12% (from our sample) or 20%, the positive predictive value (PPV) was low and negative predictive value (NPV) was high across all reference standards (Table 3). The PPV increased with higher prevalence, though remained low (16.7-26.2% at 20% prevalence).

### Accuracy of the Uganda NTLP algorithm by risk group

We evaluated the accuracy of the NTLP algorithm among children less than 5 years, CHIV, and those with SAM. For each reference standard, the sensitivity and specificity remained similar to the overall estimate with overlapping confidence intervals. Similar to the overall estimate, the specificity slightly increased without CXR (Table 4).

### Accuracy of the WHO algorithms A and B

Similar to the NTLP, the sensitivity of the WHO TDAs (children <10 years) and NPV was high but specificity and PPV was low across reference standards and TB prevalences (Table 2

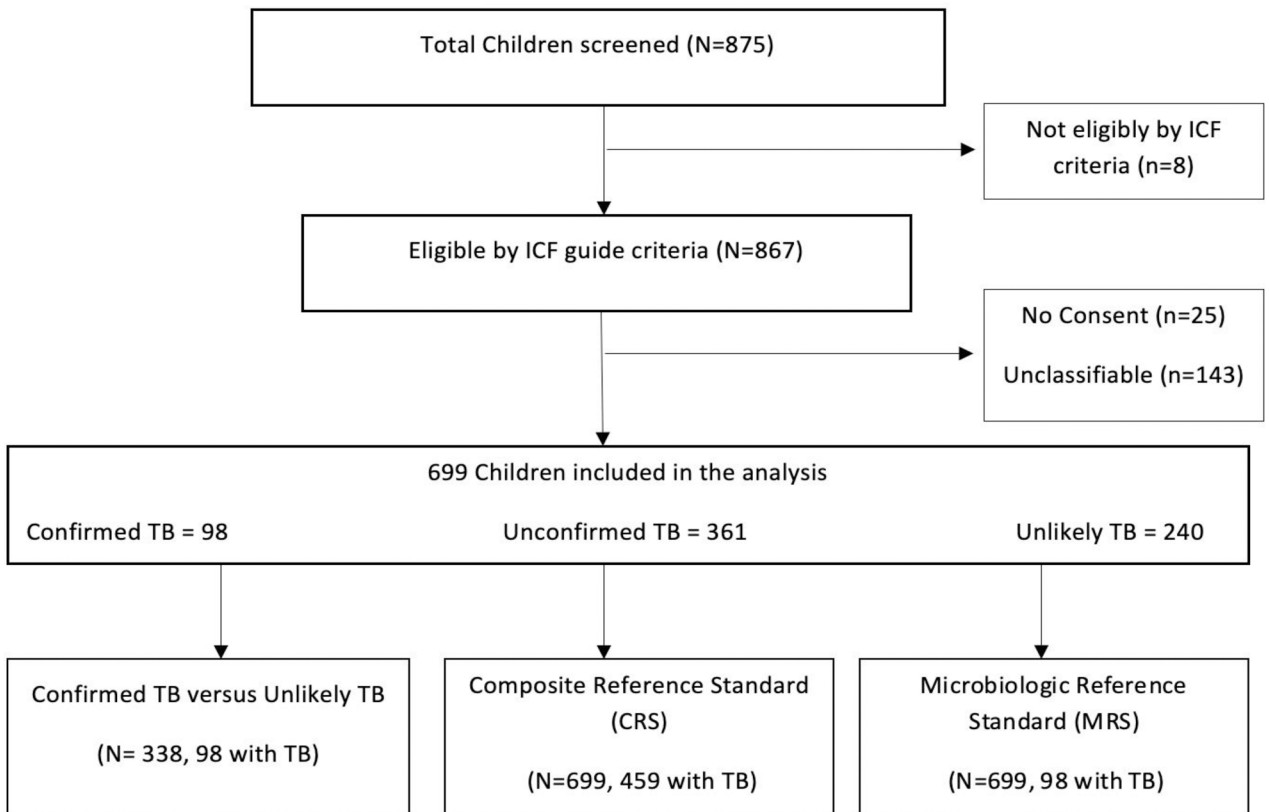

**Fig 1.** <u>Study Cohort Flow chart</u>. A schematic representation of the study population, inclusion and exclusion criteria, and final cohort of children evaluated for TB in this study.

and 3). However, in contrast, specificity varied depending on availability of CXR, with higher specificity when CXR was available for WHO algorithm A (Table 2).

## Comparison of the diagnostic accuracy of the 2017 Uganda NTLP algorithm and the 2022 WHO childhood TB diagnosis algorithms

Among children <10 years, the sensitivity and specificity of the Uganda NTLP algorithm and the WHO TDAs varied depending on the availability of Xpert and/or CXR (Table 4). We found that across the scenarios, the sensitivity between the Uganda NTLP and WHO TDAs remained similar, except WHO TDA-B had higher sensitivity compared to the Uganda NTLP algorithm when CXR was not available against the CRS only (94.2% vs. 87.5% without CXR or Xpert, difference of -7.0, [95% CI: -10.7, -3.2], p <0.01). Specificity depended on the availability of CXR; if CXR was available, WHO TDA-A was more specific than the Uganda NTLP algorithm regardless of Xpert access. However, if CXR was unavailable, specificity was lower in WHO TDA-B compared to Uganda NTLP algorithm, across reference standards (Table 5).

## Discussion

In this study, we sought to determine the accuracy of the Uganda NTLP algorithm and compare its performance to the WHO TDAs. We found that the Uganda NTLP algorithm had a high sensitivity but low specificity, under both microbiologic and composite reference standards, and the accuracy remained similar (high sensitivity and low specificity) in key

**Table 1.  Characteristics of enrolled children with presumptive pulmonary tuberculosis in Kampala, Uganda (n=699).**

| Characteristic | n (%) |
| --- | --- |
| Age Category | |
| < 1 year | 80 (11.4%) |
| 1 - < 5 years | 371 (53.1%) |
| 5 - < 10 years | 191 (27.3%) |
| 10 - < 15 years | 57 (8.2%) |
| Male sex | 373 (53.4%) |
| Has fever[1] | 560 (80.5%) |
| Has lymphadenopathy[1] | 332 (47.6%) |
| Has weight loss | 271 (38.8%) |
| Failed to gain weight[1] | 584 (84.0%) |
| Has TB contact | 399 (57.1%) |
| HIV Positive[1] | 74 (11.1%) |
| Severe acute malnutrition | 42 (6.0%) |
| Chest X-ray interpretation[1] | |
| Normal | 387 (61.5%) |
| Abnormal - Equivocal | 129 (20.5%) |
| Abnormal - Likely TB | 113 (18.0%) |
| Xpert MTB/RIF Ultra Positive | 85 (12.2%) |

1.3 missing fever, 1 missing lymphadenopathy, 4 missing Failure to gain weight, 30 missing HIV status, and 70 missing chest-xray.

risk groups (children < 5 years, CHIV, or SAM). Both the WHO TDA and the Uganda NTLP algorithms had similarly high sensitivity, but specificity was higher for the WHO TDA-A in settings with CXR available, while the Uganda NTLP algorithm had higher specificity in settings without molecular testing and/or CXR. Our findings highlight that TDAs can capture most cases with TB, but there is a need for tools to augment the specificity of these algorithms to reduce overtreatment, especially if molecular testing or CXR is not available.

The high sensitivity and high NPV of the Uganda NTLP algorithm support the goal of reducing missed TB cases, and previous work has shown that this algorithm increased TB notification among children (<15 years of age) in two Ugandan districts by 41% over an 18-month period [10]. However, our study raises concern that a large proportion of children are overtreated for TB, with low PPV across a range of TB prevalences. As children in the general population are already screened first for symptoms or risk factors prior to utilizing the Uganda NTLP algorithm, greater specificity is needed to improve its use as a diagnostic test. For the CRS, we found a small reduction in sensitivity when children with Unconfirmed TB were included, as expected given this is a heterogenous group and may include children without true TB disease. However, across reference standards, the sensitivity remained high and specificity was low, with implications for overtreatment of children. Our results were in contrast to an individual participant data (IPD) meta-analysis that showed the Uganda NTLP algorithm to have a lower sensitivity of 67% and higher specificity of 39%, [16] though there was wide heterogeneity across studies, with a range of 33-97% sensitivity and 15-85% specificity. High sensitivity and low specificity were also seen in subgroup analysis among CHIV, SAM, or below 5 years old. It may be appropriate to over-treat these higher risk groups, but further work is needed to understand the potential harms including side effects or missed alternative diagnoses.

**Table 2. Sensitivity and Specificity of the 2017 Uganda national childhood TB diagnosis algorithm (NTLP) and the 2022 WHO TB diagnosis algorithms A & B under three reference standards with or without chest x-ray (CXR). Xpert results were incorporated into the TDA classification.**

| Confirmed TB versus Unlikely TB | | |
|---|---|---|
| Algorithm[1,2] | Sensitivity % (n/N, 95% CI) | Specificity % (n/N, 95% CI) |
| *With CXR* | | |
| Uganda NTLP | 97.9 (96/98, 96.4-99.4) | 25.9 (60/231, 21.2-30.7) |
| WHO Algorithm A | 96.3 (80/83, 94.2-98.5) | 32.2 (69/214, 26.9-37.5) |
| *Without CXR* | | |
| Uganda NTLP | 97.9 (96/98, 96.4-99.4) | 28.1 (65/231, 23.2-33.0) |
| WHO Algorithm B | 97.5 (81/83, 95.8-99.3) | 15.4 (33/214, 11.3-19.5) |
| **Composite Reference Standard (CRS)** | | |
| Algorithm | Sensitivity % (n/N, 95% CI) | Specificity % (n/N, 95% CI) |
| *With CXR* | | |
| Uganda NTLP | 90.7 (402/443, 88.5-92.9) | 25.9 (60/231, 22.6-29.2) |
| WHO Algorithm A | 89.3 (360/403, 86.8-91.7) | 32.2 (69/214, 28.5-35.9) |
| *Without CXR* | | |
| Uganda NTLP | 88.4 (392/443, 86.0-90.9) | 28.1 (65/231, 24.7-31.5) |
| WHO Algorithm B | 94.2 (380/403, 92.4-96.1) | 15.4 (33/214, 12.5-18.2) |
| **Microbiologic Reference Standard (MRS)** | | |
| Algorithm | Sensitivity % (n/N, 95% CI) | Specificity % (n/N, 95% CI) |
| *With CXR* | | |
| Uganda NTLP | 97.9 (96/98, 96.8-99.0) | 17.1 (99/477, 14.3-20.0) |
| WHO Algorithm A | 96.3 (80/83, 94.9-97.8) | 20.4 (109/534, 17.3-23.5) |
| *Without CXR* | | |
| Uganda NTLP | 97.9 (96/98, 96.8-99.0) | 19.7 (114/576, 16.7-22.8) |
| WHO Algorithm B | 97.5 (81/83, 96.3-98.8) | 10.1 (54/534, 7.7-12.4) |

CXR, chest x-ray; n/N, Proportion of children with the outcome relative to the total population in that particular cell. WHO Algorithm A, settings with chest X-ray; WHO Algorithm B, settings without chest X-ray.

1. Algorithms were applied to the age group they were indicated for: the NTLP for children <15 years, and <10 years for the WHO TDAs.

2. Xpert Ultra results included in the algorithms.

The accuracy of the WHO TDAs was consistent with the IPD meta-analysis that found the pooled sensitivity and specificity at 80% and 30% respectively [16]. Our higher sensitivity may be explained by the fact that children in our study were first screened using a more sensitive ICF, as similarly seen among adults in Kampala where there was a 6% increase in TB case notifications as a result of ICF implementation [17]. Comparing the Uganda NTLP algorithm and WHO TDAs, the sensitivity remained high regardless of Xpert, reflecting that these have been developed in particular for empiric treatment when microbiological testing is not available or frequently negative, given paucibacillary disease in children. The Uganda NTLP algorithm's accuracy remained stable with or without CXR, unlike the WHO TDA's specificity that dropped in scenarios where CXR was unavailable. A possible reason for this is how each algorithm is scored. For the Uganda NTLP algorithm, the presence or absence of specific signs or symptoms are equally weighted, and so more children may be diagnosed with TB for example regardless of CXR findings. In contrast, the WHO TDAs assign points that differentially weight signs and symptoms, including specific CXR findings. Moreover, the assigned points are different whether CXR is present or not, and the internal validation data showed that the

**Table 3. Positive and Negative Predictive Values of the 2017 Uganda national childhood TB diagnosis algorithm (NTLP) and the 2022 WHO TB diagnosis algorithms A & B under three reference standards with or without chest x-ray (CXR) at different prevalence levels (5%, 12%, and 20%).**

| | Prevalence (5%)[1] | | Prevalence (12%)[1] | | Prevalence (20%)[1] | |
|---|---|---|---|---|---|---|
| Scenario | PPV % | NPV % | PPV% | NPV % | PPV % | NPV % |
| Confirmed Vs. Unlikely | | | | | | |
| NTLP-Xpert & CXR | 6.5% | 99.5% | 15.2% | 98.9% | 24.8% | 98.0% |
| NTLP-Xpert only | 6.6% | 99.6% | 15.6% | 99.0% | 25.4% | 98.2% |
| NTLP-CXR only | 6.1% | 98.2% | 14.4% | 95.6% | 23.5% | 92.3% |
| NTLP Without Xpert + CXR | 6.1% | 98.0% | 14.4% | 95.0% | 23.6% | 91.2% |
| WHO-A | 6.9% | 99.4% | 16.2% | 98.4% | 26.2% | 97.2% |
| WHO-B | 5.7% | 99.4% | 13.6% | 97.9% | 22.3% | 96.2% |
| Composite reference standard (CRS) | | | | | | |
| NTLP-Xpert & CXR | 6.0% | 98.1% | 14.3% | 95.3% | 16.7% | 98.0% |
| NTLP-Xpert only | 6.0% | 97.8% | 14.3% | 94.7% | 23.5% | 90.7% |
| NTLP-CXR only | 5.9% | 97.8% | 14.1% | 94.6% | 23.1% | 90.6% |
| NTLP Without Xpert + CXR | 5.9% | 97.5% | 14.1% | 93.8% | 23.1% | 89.3% |
| WHO-A | 6.4% | 98.2% | 15.2% | 95.6% | 24.7% | 92.3% |
| WHO-B | 5.5% | 98.0% | 13.2% | 95.2% | 21.8% | 91.5% |
| Microbiologic reference standard (MRS) | | | | | | |
| NTLP-Xpert & CXR | 5.8% | 99.3% | 13.8% | 98.4% | 22.8% | 97.1% |
| NTLP-Xpert only | 6.0% | 99.4% | 14.2% | 98.6% | 23.3% | 97.4% |
| NTLP-CXR only | 5.4% | 97.4% | 13.0% | 93.6% | 21.6% | 88.8% |
| NTLP Without Xpert + CXR | 5.5% | 97.2% | 13.1% | 93.1% | 21.7% | 88.0% |
| WHO-A | 5.9% | 99.0% | 14.1% | 97.6% | 23.2% | 95.7% |
| WHO-B | 5.4% | 98.7% | 12.9% | 96.8% | 21.3% | 94.3% |

1. Based on the TB prevalence, sensitivity and specificity of the corresponding scenario.

PPV, Positive predictive value; NPV, Negative predictive value; CXR, Chest x-ray; WHO-A, settings with chest x-ray; WHO-B, settings without chest x-ray.

model with CXR had greater specificity [16]. From an implementation perspective, the WHO TDAs and algorithms with a similar approach may be most valuable in settings with both Xpert and CXR available, but those are facilities where providers may have more experience with diagnosing TB. Although the Uganda NTLP algorithm had lower specificity than WHO TDA-A, this was higher than TDA-B and it remained stable with or without CXR; it therefore may be preferable in lower-level health facilities which may not have molecular or radiological testing facilities. Policy makers will need to balance the benefits of detecting more cases with the costs of over-treatment and expanding access to molecular testing with CXR to improve specificity.

Our study adds much needed data on the accuracy of the Uganda NTLP algorithm and its comparison to the WHO TDAs. Our data was not used to develop the WHO TDAs, and so will be valuable for the external validation. We further present the accuracy results based on availability of CXR and Xpert to guide how these algorithms may perform in different types of settings. Additionally, our TB prevalence (12%) was similar to what was found in a study done at two Ugandan referral hospitals [18], but we further described PPV and NPV estimates in settings that may have lower or higher TB prevalence. Our results show that the benefits of the TDAs may be greater in high TB prevalence settings. However, our study had limitations. The CRS is challenging because Unconfirmed TB can be greatly influenced by provider treatment initiation, which in turn may have been informed by the NTLP algorithm given it is implemented at the facility. Therefore, we will have high incorporation bias as the CRS

**Table 4. Sensitivity and specificity of the 2017 Uganda national childhood TB diagnosis algorithm (NTLP) by risk group (Age < 5 years, SAM, and HIV) under three reference standards with or without chest x-ray (CXR) for children <15 years old.**

| Group | Sensitivity % (n/N, 95% CI) | Specificity % (n/N, 95% CI) |
|---|---|---|
| | **Confirmed TB versus Unlikely TB** | |
| Age < 5 years | | |
| With CXR | 98.3 (60/61, 96.5-100.1) | 29.4 (40/136, 23.0-35.7) |
| Without CXR | 98.3 (60/61, 96.5-100.1) | 32.3 (44/136, 25.8-38.8) |
| SAM | | |
| With CXR | 97.9 (96/98, 96.4-99.4) | 25.9 (60/231, 21.2-30.7) |
| Without CXR | 97.9 (96/98, 96.4-99.4) | 28.1 (65/231, 23.0-33.0) |
| HIV | | |
| With CXR | 97.8 (91/93, 96.2-99.4) | 25.9 (60/231, 21.2-30.7) |
| Without CXR | 97.8 (91/93, 96.2-99.4) | 28.1 (65/231, 23.0-33.0) |
| | **Composite Reference Standard (CRS)** | |
| Age < 5 years | | |
| With CXR | 89.9 (267/297, 87.0-92.7) | 29.4 (40/136, 25.1-33.7) |
| Without CXR | 88.2 (262/297, 85.1-91.2) | 32.3 (44/231, 27.9-36.7) |
| SAM | | |
| With CXR | 90.7 (402/443, 88.5-92.9) | 25.9 (60/231, 22.6-29.3) |
| Without CXR | 88.4 (392/443, 86.0-90.9) | 28.1 (65/231, 24.7-31.5) |
| HIV | | |
| With CXR | 90.6 (397/438, 88.4-92.8) | 25.9 (60/231, 22.6-29.3) |
| Without CXR | 88.3 (387/438, 85.9-90.7) | 28.1 (65/231, 24.7-31.5) |
| | **Microbiologic Reference Standard (MRS)** | |
| Age < 5 years | | |
| With CXR | 98.3 (60/61, 97.1-99.5) | 18.5 (69/308, 14.8-22.2) |
| Without CXR | 98.3 (60/61, 97.1-99.5) | 20.9 (78/372, 17.1-24.8) |
| SAM | | |
| With CXR | 97.9 (96/98, 96.8-99.0) | 17.1 (99/576, 14.3-20.0) |
| Without CXR | 97.6 (96/98, 96.8-99.0) | 19.7 (114/576, 16.7-22.8) |
| HIV | | |
| With CXR | 97.8 (91/93, 96.7-98.9) | 17.1 (99/576, 14.3-20.0) |
| Without CXR | 97.8 (91/93, 96.9-98.9) | 19.7 (114/576, 16.7-22.8) |

CXR, Chest x-ray; SAM, Severe Acute Malnutrition; HIV, Human Immunodeficiency virus.

includes both Xpert, CXR and clinical decision to treat. Another limitation is that we looked at urban children with presumptive TB and therefore our results may not generalize to rural or migratory populations. Additionally, the assessment was done by study staff focused on the evaluation of TB and all children provided respiratory samples, which could reduce generalizability. The WHO TDAs are designed for use at lower levels of health care, meaning that use of data from a tertiary level may introduce bias. Although the study site was a tertiary hospital, many of the children were referred from lower-level health facilities. Another limitation was our retrospective design; the WHO TDAs have additional components (two weeks follow up for low-risk children, assessment of danger signs) that we were unable to include in our analysis, and may have biased accuracy estimates. However, our analysis allowed comparison on the performance of the WHO TDAs if implemented similarly to the NTLP for all children

**Table 5. Comparison of 2017 Uganda national childhood TB diagnosis algorithm (NTLP) against 2022 WHO childhood TB treatment decision algorithms among children <10 years old with and without Xpert testing or Chest X-ray imaging.**

| Scenario | NTLP | | WHO Algorithm A/B[1] | | | |
|---|---|---|---|---|---|---|
| | Sensitivity %, (95%CI) | Specificity %, (95%CI) | Sensitivity %, (95%CI) | Specificity %, (95%CI) | Difference in Sensitivity (NTLP vs. WHO) % (95% CI), p-value | Difference in Specificity (NTLP vs. WHO) % (95% CI), p-value |
| Confirmed TB Vs. Unlikely TB | | | | | | |
| Xpert + CXR | 97.6 (95.9, 99.3) | 25.7(20.7, 30.7) | 96.3 (94.2, 98.5) | 32.2 (26.9, 37.5) | 1.3 (-2.3, 4.5), 1.00 | -6.8 (-11.9, -1.7), 0.01 |
| Xpert | 97.6 (95.9, 99.3) | 28.1 (22.9, 33.2) | 97.5 (95.8, 99.3) | 15.4 (11.3, 19.5) | 0 (-1.2, 1.2), 1.00 | 12.1 (5.3, 19.0), <0.01 |
| CXR | 93.7 (90.9, 96.5) | 25.7 (20.6, 30.7) | 96.3 (94.2, 98.5) | 32.2 (26.9, 37.5) | -2.5 (-3.7, 8.8), 0.62 | -6.8 (-11.9, -1.7), <0.01 |
| None | 91.2 (88.0, 94.5) | 28.1 (22.9, 33.2) | 97.5 (95.8, 99.3) | 15.4 (11.3, 19.5) | -6.4 (-0.3, 13.1), 0.06 | 12.1 (5.3, 19.0), <0.01 |
| Composite reference standard (CRS) | | | | | | |
| Xpert + CXR | 90.6 (88.3, 92.9) | 25.7 (22.2, 29.1) | 89.3 (86.8, 91.7) | 32.2 (28.5, 35.9) | 1.0 (-1.9, 4.0), 0.58 | -6.8 (-11.9, -1.7), <0.01 |
| Xpert | 88.9 (86.5, 91.4) | 28.1 (24.5, 31.6) | 94.2 (92.4, 96.1) | 15.4 (12.5, 18.2) | -5.6 (2.1, 9.2), <0.01 | 12.1 (5.3, 19.0), <0.01 |
| CXR | 89.8 (87.4, 92.2) | 25.7 (22.2, 29.1) | 89.3 (86.8, 91.7) | 32.2 (28.5, 35.9) | 0 (-2.9, 3.4), 1.00 | -6.8 (-11.9, -1.7), <0.01 |
| None | 87.5 (84.9, 90.2) | 28.1 (24.5, 31.6) | 94.2 (92.4, 96.1) | 15.4 (12.5, 18.2) | -7.0 (-10.7, -3.2), <0.01 | 12.1 (5.3, 19.0), <0.01 |
| Microbiologic reference standard (MRS) | | | | | | |
| Xpert + CXR | 97.6 (96.4, 98.8) | 16.8 (13.9, 19.8) | 96.3 (94.9, 97.8) | 20.4 (17.2, 23.5) | 1.2 (-2.3, 4.7), 1.00 | -3.3 (-0.3, 6.2), 0.02 |
| Xpert | 97.6 (96.4, 98.8) | 19.1 (16.0, 22.2) | 97.5 (96.3, 98.8) | 10.1 (7.7, 12.4) | 0 (-1.2, 1.2), 1.00 | 9.1 (5.4, 12.9), <0.01 |
| CXR | 93.7 (91.8, 95.6) | 16.8 (13.9, 19.8) | 96.3 (94.9, 97.8) | 20.4 (17.2, 23.5) | -2.5 (-3.7, 8.8), 0.62 | -3.3 (-0.3, 6.2), 0.02 |
| None | 91.2 (89.0, 93.4) | 19.1 (16.0, 22.2) | 97.5 (96.3, 98.8) | 10.1 (7.7, 12.4) | -6.4 (-0.3, 13.1), 0.06 | 9.1 (5.4, 12.9), <0.01 |

Xpert, Xpert MTB/RIF Ultra; CXR, Chest x-ray.

1. Algorithm A used in scenarios with CXR, Algorithm B in scenarios without CXR.

with presumptive TB. Also, our analysis was restricted to only children who fulfilled the ICF criteria; however, only eight children were excluded as a result of this requirement.

In summary, our study shows that if implemented in its current form, the Uganda NTLP algorithm and WHO TDAs have high sensitivity to capture most cases of childhood TB, but may result in overdiagnosis and overtreatment of children without TB. Our study further highlights a potential limitation of implementing the WHO TDAs over the Uganda NTLP algorithm in primary care clinics without access to CXR or molecular testing. As new TB diagnostics emerge, it will be important to integrate them into TDAs to assess incremental changes in accuracy.

## Supporting information

**S1 Appendix: Uganda NTLP diagnostic algorithm for childhood TB.** A clinical decision tool designed by the Uganda national Tuberculosis and Leprosy Program (NTLP) to aid healthcare providers in diagnosing and managing TB in children.
(TIF)

**S2 Appendix: World Health Organization Treatment Decision Algorithm-A.** Treatment decision algorithms (TDAs) revised in 2022 by the World Health Organisation to improve TB diagnosis in children in settings with chest x-ray.
(TIF)

**S3 Appendix: World Health Organization Treatment Decision Algorithm-B.** Treatment decision algorithms (TDAs) revised in 2022 by the World Health Organisation to improve TB diagnosis in children in settings without chest x-ray.
(TIF)

**S4 Appendix: Standards for Reporting Diagnostic Accuracy (STARD) guidelines.** Guidelines followed by this study to ensure that we provide sufficient details on methodology, execution, and results, allowing for a comprehensive assessment of our test's validity and applicability.
(TIF)

AcknowledgmentWe thank the patients, families, and staff at Mulago National Referral Hospital, Kampala Capital City Authority Clinics, and Infectious Diseases Institute (IDI) clinics.

## Author contributions

**Conceptualization:** Peter J Kitonsa, Eric Wobudeya, Devan Jaganath.

**Data curation:** Peter J Kitonsa, Jascent Nakafeero, Gertrude Nanyonga, Emma Kiconco, Deus Atwiine, Robert Castro, Ernest A Oumo.

**Formal analysis:** Peter J Kitonsa, Annet Nalutaaya, Robert Castro, Devan Jaganath.

**Funding acquisition:** Adithya Cattamanchi, Devan Jaganath.

**Investigation:** Peter Wambi, Jascent Nakafeero, Gertrude Nanyonga, Emma Kiconco, Deus Atwiine, Ernest A Oumo, Adithya Cattamanchi, Eric Wobudeya, Devan Jaganath.

**Methodology:** Peter J Kitonsa, Bernard Kikaire, Ezekiel Mupere, Adithya Cattamanchi, Eric Wobudeya, Devan Jaganath.

**Project administration:** Peter Wambi, Deus Atwiine, Mary N Mudiope, Adithya Cattamanchi, Eric Wobudeya, Devan Jaganath.

**Resources:** Robert Castro, Adithya Cattamanchi, Eric Wobudeya, Devan Jaganath.

**Supervision:** Bernard Kikaire, Peter Wambi, Jascent Nakafeero, Hellen T Aanyu, Mary N Mudiope, Ezekiel Mupere, Moorine P Sekadde, Swomitra Mohanty, Adithya Cattamanchi, Eric Wobudeya, Devan Jaganath.

**Visualization:** Devan Jaganath.

**Writing – original draft:** Peter J Kitonsa.

**Writing – review & editing:** Peter J Kitonsa, Bernard Kikaire, Peter Wambi, Hellen T Aanyu, Mary N Mudiope, Ezekiel Mupere, Moorine P Sekadde, Swomitra Mohanty, Adithya Cattamanchi, Eric Wobudeya, Devan Jaganath.

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
