## [Decision Letter · Decision Letter 0]

30 Dec 2024

PGPH-D-24-02642

The Accuracy of the Uganda National Tuberculosis and Leprosy Program algorithm and the World Health Organisation treatment decision algorithms for childhood tuberculosis: A retrospective analysis

Dear Dr. Kitonsa,

Thank you for submitting your manuscript to PLOS Global Public Health. After careful consideration, we feel that it has merit but does not fully meet PLOS Global Public Health’s publication criteria as it currently stands. Therefore, we invite you to submit a revised version of the manuscript that addresses the points raised during the review process.

We look forward to receiving your revised manuscript.

Kind regards,

Joseph Baruch Baluku, MMed

Academic Editor

Journal Requirements:

**Please only choose the relevant sentences from below**

1. Please clarify all sources of funding (financial or material support) for your study. List the grants (with grant number) or organizations (with url) that supported your study, including funding received from your institution. 

2. State the initials, alongside each funding source, of each author to receive each grant.

3. State what role the funders took in the study. If the funders had no role in your study, please state: “The funders had no role in study design, data collection and analysis, decision to publish, or preparation of the manuscript.”

4. If any authors received a salary from any of your funders, please state which authors and which funders.

2. We note that your Data Availability Statement is currently as follows: "Data set uploaded as supplementary material”

3. Please provide separate figure files in .tif or .eps format.

4. Please provide an Author Summary. This should appear in your manuscript between the Abstract (if applicable) and the Introduction, and should be 150–200 words long. The aim should be to make your findings accessible to a wide audience that includes both scientists and non-scientists. Sample summaries can be found on our website under Submission Guidelines:

https://journals.plos.org/globalpublichealth/s/submission-guidelines#loc-parts-of-a-submission

5. Tables should not be uploaded as individual files. Please remove these files and include the Tables in your manuscript file as editable, cell-based objects. For more information about how to format tables, see our guidelines: 

https://journals.plos.org/globalpublichealth/s/tables

6. We have noticed that you have uploaded Supporting Information files, but you have not included a list of legends. Please add a full list of legends for your Supporting Information files after the references list. 

Additional Editor Comments (if provided):

Reviewers' comments:

Reviewer's Responses to Questions

**Comments to the Author**

1. Does this manuscript meet PLOS Global Public Health’s publication criteria ? Is the manuscript technically sound, and do the data support the conclusions? The manuscript must describe methodologically and ethically rigorous research with conclusions that are appropriately drawn based on the data presented.

Reviewer #1: No

Reviewer #2: Yes

2. Has the statistical analysis been performed appropriately and rigorously?

Reviewer #1: No

Reviewer #2: Yes

3. Have the authors made all data underlying the findings in their manuscript fully available (please refer to the Data Availability Statement at the start of the manuscript PDF file)?

Reviewer #1: No

Reviewer #2: Yes

4. Is the manuscript presented in an intelligible fashion and written in standard English?

Reviewer #1: Yes

Reviewer #2: Yes

5. Review Comments to the Author

Reviewer #1: RE: PLOS Global Health 20241121

Thanks for the opportunity to review PLOS Global Health 20241121. This paper attempts to add more knowledge to the knowledge base regards the use of Treatment Diagnostic Algorithms for Pediatric TB. Especially given that the current WHO Treatment Diagnostic Algorithm has a conditional recommendation and there is need for more evidence.

However, I have some comments, and they are written line by line below:

Generally, the authors should follow the guidance for writing Diagnostic Accuracy Tests, Bossuyt PM, Reitsma JB, Bruns DE, Gatsonis CA, Glasziou PP, Irwig L, LijmerJG Moher D, Rennie D, de Vet HCW, Kressel HY, Rifai N, Golub RM, Altman DG, Hooft L, Korevaar DA, Cohen JF, For the STARD Group. STARD 2015: An Updated List of Essential Items for Reporting Diagnostic Accuracy Studies.

Line 1: It is important to clarify that this study is about the diagnostic accuracy of the Uganda National Childhood TB Algorithm and the WHO treatment algorithm for childhood Tuberculosis. Adding the words diagnostic and childhood TB helps with clarity.

Line 28: I would put a full stop after challenge. Then begin the next sentence with this has led, this helps with clarity. The sentence is shorter and easier to read.

Line 32: Add that the Uganda NTLP algorithm is for diagnosing childhood TB.

INTRODUCTION

Line 53-56: Could you give a mention to other samples that can be used to diagnose TB in children i.e stool. Which most people think would be easier to obtain.

Line 59: At the end of delays in treatment, there is need for a reference or citation.

Line 73: There is a need for a reference or citation.

MATERIALS AND METHODS

It may be important to start the methods, by clarifying what the study design is or was.

Line 81: You say the primary data. Was this a secondary data analysis. If yes, what was the primary use of the data.

Line 92-100: The criterion for the inclusion is not clear, because there are two criteria given one is the ICF and there is another described between line 92-96. Then line 96-100.

Line 116-line 120: could have been written better for clarity, please consider re-writing it.

Line 122-27: Please consider explain the WHO TDA better and also add both it and the Uganda National TB algorithm to the annexes.

RESULTS

The results could be better reported, there are other dimensions of diagnostic accuracy, apart from the sensitivity and specificity. These include negative predictive values, positive predictive values and there is also needed to provide raw data for the true positives, false positives, and true negatives and false negatives. Then you could also calculate the overall diagnostic accuracy of the test.

Line 188-190: Please write the results and refer to the Tables for further details not the other way around.

DISCUSSION

The discussion for diagnostic accuracy seeks to answer a few questions.

1. Can we use this test?

2. If we can use this test, when along the diagnostic process can we use it, is it for screening or diagnosis

Alternatively, you raised a few questions in lines 69-70. “Moreover, it is unclear if the new WHO algorithm should be utilized in Uganda, or the current Uganda NTLP algorithm should be continued or updated”. This paper doesn’t answer this question.

The Negative Predictive Value (NPV) and Positive Predictive Value (PPV) are sensitive to the prevalence of disease, so it is important to find out what happens to these values at 5%, 10% and 15% or 20%. This will further guide when these algorithms can be used.

Reviewer #2: Thank you for the opportunity to review this paper comparing the accuracy of the Ugandan NTLP algorithm and the new WHO treatment decision algorithms for the diagnosis of pulmonary TB in children. This is an important topic as WHO has called for external validation of the new treatment decision algorithms, considering that these have only been internally validated.

Comments

Line 36: When referring to the NIH definition, it may be useful to in to indicate the main classifications (confirmed, unconfirmed, unlikely TB)

Line 45: For enhanced understanding it may be good to provide a definition of high risk children (high risk of rapid disease progression/severe disease/something else?)

Line 61: there is a reference to the 2021 Global Tuberculosis report, however, 5-year progress against the 2018 UN HLM targets was presented in the 2023 report and the % was 71% (the 41% was an interim figure after only 3 years so this should not be used in this context). It may be better to use treatment coverage as an indicator here rather than progress against the UN HLM target. In 2023 (as reported in the 2024 Global TB report) treatment coverage was 55% for the age group <15 years (but only 48% in the age group below 5 years).

Line 67: please clarify that the TDAs are included in the WHO operational handbook and that the recommendation in the guidelines is on TDAs in general (not for any algorithm in particular).

Line 67-69: the main issue to note is that the TDAs in the WHO operational handbook have only been internally validated and WHO has called for external validation of the algorithms.

Study population: it will also be important to note (in the discussion) that the WHO TDAs are designed for use at lower levels of the health care system, and conducting a retrospective analysis of tertiary level data may introduce bias (this needs to be included in the discussion under limitations).

Line 114: it may be good to include the NTLP algorithm/decision tree and the WHO TDAs as images to clarify the steps if possible.

Lines 122-127: an important omission in the description of the WHO TDAs is the steps that are included before molecular testing is done: these include danger signs and relevant urgent management and also an assessment of risk of rapid disease progression, whereby children at high risk of rapid disease progression (those aged <2 years, those with HIV infection or those with SAM) immediately continue through the algorithms, but children at low risk of rapid disease progression are first treated for alternative diagnoses and only return for the next steps if they have not improved after this treatment. In a retrospective study, it will be very difficult to correctly evaluate this step as this data may not be available. This is a major limitation of this study and this needs to be acknowledged in the discussion. It is not clear from the description of the methodology which steps of the WHO TDAs were taken into account, this will need to be further clarified. It seems that only the results of molecular testing and the scoring sections were considered.

Line 149: for the overall…. It seems there is a word missing

Line 228-229: this statement suggests that the only part that was taken into account was the scoring section of the WHO TDAs, but not the additional steps (danger signs and risk assessment) that were added in an attempt to improve accuracy (notably specificity). It suggests therefore that the retrospective methodology is not adequate to assess these algorithms, and a prospective design would be needed to evaluate the algorithms comprehensively. It will be important to highlight this as a limitation in the discussion.

Lines 239-240: it may be good to explain why the score in the WHO algorithm differs – and reference the Gunasekera paper here (currently ref 12)

From line 252 on limitations: here the additional limitations mentioned above (mainly related to disregarding the additional steps in the WHO algorithms) need to be included. It may be good to mention that an operational research study is currently being implemented to prospectively evaluate the WHO TDAs in various settings in Uganda. In addition, it will be important to check and if so, to state, if a subset of the data reviewed here in this dataset was used for the development of the WHOP TDAs (see also the Gunasekera paper).

Lines 256-257: in addition the algorithms (both the NTLP and WHO ones) are designed to be used at PHC level, where the prevalence of TB may be lower and where children generally present with less severe forms of TB, and where capacity to conduct Xpert testing/specimen collection and CXR may be limited and thus overdiagnosis may be an important issue in these settings.

From line 261: it will be important to flag here the limitations of the retrospective methodology as mentioned before. Capacity building and regular mentorship will be other important aspects to take into consideration.

References: need substantial editing as many authors/organizations are not reflected correctly.

6. PLOS authors have the option to publish the peer review history of their article (what does this mean? ). If published, this will include your full peer review and any attached files.

**Do you want your identity to be public for this peer review?** For information about this choice, including consent withdrawal, please see our Privacy Policy .

Reviewer #1: **Yes:**

Reviewer #2: No

---

## [Editor Report · Decision Letter 1]

25 Feb 2025

The Accuracy of the Uganda National Tuberculosis and Leprosy Program algorithm and the World Health Organisation treatment decision algorithms for childhood tuberculosis: A retrospective analysis

PGPH-D-24-02642R1

Dear Dr Kitonsa,

We are pleased to inform you that your manuscript 'The Accuracy of the Uganda National Tuberculosis and Leprosy Program algorithm and the World Health Organisation treatment decision algorithms for childhood tuberculosis: A retrospective analysis' has been provisionally accepted for publication in PLOS Global Public Health.

Best regards,

Joseph Baruch Baluku, MMed

Academic Editor